# Visualizing high-dimensional trajectories on the loss-landscape of ANNs

**Stefan Horoi**[*]
Dept. of Math. and Stat.
Univ. de Montréal ; Mila
Montreal, QC, Canada
stefan.horoi@umontreal.ca

**Jessie Huang**[*]
Dept. of Comp. Sci.
Yale University
New Haven, CT, USA
jiexi.huang@yale.edu

**Guy Wolf**[†]
Dept. of Math. and Stat.
Univ. de Montréal ; Mila
Montreal, QC, Canada
guy.wolf@umontreal.ca

**Smita Krishnaswamy**[†]
Depts. of Gene. & Comp. Sci.
Yale University
New Haven, CT, USA
smita.krishnaswamy@yale.edu

## Abstract

Training artificial neural networks requires the optimization of highly non-convex loss functions. Throughout the years, the scientific community has developed an extensive set of tools and architectures that render this optimization task tractable and a general intuition has been developed for choosing hyper parameters that help the models reach minima that generalize well to unseen data. However, for the most part, the difference in trainability in between architectures, tasks and even the gap in network generalization abilities still remain unexplained. Visualization tools have played a key role in uncovering key geometric characteristics of the loss-landscape of ANNs and how they impact trainability and generalization capabilities. However, most visualizations methods proposed so far have been relatively limited in their capabilities since they are of linear nature and only capture features in a limited number of dimensions. We propose the use of the modern dimensionality reduction method PHATE which represents the SOTA in terms of capturing both global and local structures of high-dimensional data. We apply this method to visualize the loss landscape during and after training. Our visualizations reveal differences in training trajectories and generalization capabilities when used to make comparisons between optimization methods, initializations, architectures, and datasets. Given this success we anticipate this method to be used in making informed choices about these aspects of neural networks.

## 1 Introduction

Artificial neural networks (ANNs) have been successfully used to solve a number of complex tasks in a diverse array of domains. However, questions such as why ANNs favor generalization over memorization and why they find good minima even with intricate loss functions still remain largely unanswered. One promising research direction for answering them is to look at the loss-landscape of deep learning models. Recent work has approached this task by proposing various visualization methods. In most papers, loss functions and their level lines have been visualized via linear methods like PCA. In some case, this approach proved effective in uncovering underlying structures in the

---

[*]Equal contribution. [†]Equal senior-author contribution; corresponding authors.

Deep Learning through Information Geometry (DL-IG) Workshop
34th Conference on Neural Information Processing Systems (NeurIPS 2020), Vancouver, Canada.

loss-landscape and linking them to ANN generalization or architecture [1, 2]. However, these methods have two major key drawbacks: **(1)** they only choose directions that are linear combinations of parameter axes while the loss landscape itself is highly nonlinear, and **(2)** they choose only two among millions of axes to visualize and ignore all others.

In this work, we utilize and adapt the PHATE dimensionality reduction method [3], which relies on diffusion-based manifold learning, to study ANN loss landscapes by visualizing the evolution of network weights during training in low dimensions. In general, unlike PCA, visualizations like PHATE [3] are specifically designed to squeeze as much variability as possible into two dimensions, and thus provide an advantage over previous approaches. In particular our choice of using PHATE over other popular methods, such as tSNE [4], is due to its ability to retain both global and local structures of data, and in particular to keep intact the training trajectories that are traversed through during gradient descent. Indeed, During training, the high-dimensional neural networks weights change significantly while remaining on a connected manifold defined by the support of viable configurations which we refer to when discussing the geometry of the loss landscape. We show that PHATE is suitable to track such continuous weight trajectories, as opposed to tSNE or UMAP that tend to shatter them. Moreover, our approach provides general view of relevant geometric patterns that emerge in the high-dimensional parameter space, providing insights regarding the properties of ANN training and reflecting on their impact on the loss landscape.

## 2 Related work

Loss landscape visualization methods in literature are mostly based on linear interpolations, either 1D interpolations between the initial and final network parameters [5], or scanning a 2D grid around the optimum [2]. There is growing interest in uncovering the high-dimensional and complex geometric and topological characteristics of loss landscapes beyond linear paths. Nonlinear pathways with consistently low losses connecting distinct minima were found in [6], suggesting the existence of connected low loss manifolds. Such high-dimensional manifolds are beyond the capability of linear methods. Even in standard applications of the linear methods one inevitably asks if the millions of unseen directions do not hide critical features of the landscape, and if the visualized linear path is relevant to what happened during training. We suggest the use of dimensionality reduction techniques to study the complex loss-landscape of ANNs. Such techniques have been extensively used recently to successfully study the internal learned representations of ANNs and highlight their complex geometric structures and intrinsic dimensionality [7, 8, 9, 10, 11]. Here we further advance this line of work to provide new applications and insights in the study of ANN optimization and generalization.

## 3 Background: PHATE dimensionality reduction & visualization

Given a data matrix $\mathbf{N}$, PHATE first computes the pairwise similarity matrix $\mathbf{A}$ (e.g. via a Gaussian kernel), then row-normalize $\mathbf{A}$ to obtain the diffusion operator $\mathbf{P}$, a row-stochastic Markov transition matrix where $\mathbf{P}_{i,j}$ denotes the probability of moving from the $i$-th to the $j$-th data point in one time step. Powers of the matrix $\mathbf{P}^t$ represents moving along the Markov chain for $t$ steps, where $t$ is selected automatically as the knee point of the Von Neumann Entropy of the diffusion operator. To enable dimensionality reduction while retaining diffusion geometry information from the operator, PHATE leverages *information geometry* to define a pairwise *potential distance* as an M-divergence $\mathbf{ID}_{i,j} = \|\log P_{i,:} - \log P_{j,:}\|_2$ between corresponding $t$-step diffusion probability distributions of the two points, which provides a *global context* to each data point. The resulting information distance matrix $\mathbf{ID}$ is finally embedded into a visualizable low dimensional (2D or 3D) space by metric multidimensional scaling (MDS), thereby squeezing the intrinsic geometric information to calculate the final 2D or 3D embeddings of the data. For further details, see Moon et al. [3].

## 4 Methods

The dimensionality of the space renders the task of completely visualizing the loss-landscape of ANNs virtually impossible. By gathering locally connected loss-landscape patches throughout training and around minima found via gradient descent, we hope to reconstruct manifolds on which these trajectories lie. Despite the high dimensions of the parameter space $\Theta$, we hypothesize that these trajectories and manifolds are intrinsically low dimensional and thus PHATE, which reduced dimensionality via diffusion-based manifold learning and information geometry, will help facilitate their visualization in two specific settings: 1. To characterize the region of the loss-landscape surrounding minima; 2. To simultaneously visualize multiple training trajectories corresponding to

different parameter initializations and optimizers in order to make training choices. In setting **(1)** we use the following "jump and retrain" experiment. Given a trained ANN and the final parameters $\theta_o$,

- For combinations of $seed \in \{0, 1, 2, 3, 4\}$ and $step\_size \in \{0.25, 0.5, 0.75, 1\}$:
  1. Choose the random $v_{seed}$ in $\Theta$ and filter normalize it as proposed in [2] to obtain $\overline{v}_{seed}$;
  2. Set the network parameters at each jump initialization to be $\theta_{\text{jump-init}} = \theta_o + step\_size \cdot \overline{v}_{seed}$;
  3. Retrain for 50 epochs on the original training set and record $\theta$ at the end of each epoch.
- Apply PHATE to the data matrix $[\theta_{seed, step\_size, epoch}]$ and visualize the embeddings.

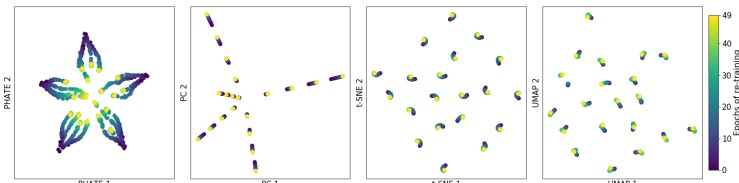

Figure 1: 2D embeddings of the jump and retrain experiment results for a WResNet28-2 network using the modern dimensionality reduction techniques PHATE, t-SNE, UMAP with cosine distance and linear PCA. See Appendix A for higher resolution and experiment ran on tree-like dataset.

In setting **(2)**, we simultaneously visualize multiple training trajectories corresponding to different parameter initializations and optimizers. In both settings, PHATE has allowed us to bypass the key drawbacks of previously proposed linear interpolation methods by: 1. Capturing variance in data from **all** relevant dimensions and embed it in a low-dimensional space; and 2. Preserving high-dimensional trajectories and global relationships to a greater extent than any other state-of-the-art dimensionality reduction technique. Figure 1 demonstrates this by showing a comparison of multiple such techniques, namely PHATE, PCA, t-SNE [4] and UMAP [12], and how they each embed the results of a jump and retrain experiment. While some trajectory-like structure is visible in all low-dimensional embeddings, only PHATE properly captures intra-trajectory variance and global relationships in between trajectories while t-SNE and UMAP have a tendency to cluster points that are close in parameter space and disregard the global structure of the data.

## 5   Generalization vs. Memorization

Arguably one of the most important and general questions in machine learning at the moment is: by looking only at a given trained model and its training data, can we predict how well the model will perform on unseen data? Previous work offers a partial and vague answer to this question by linking the "flatness" of neural network optima to the extent to which the network is able to generalize. This idea was initially presented in [13] and has recently been the focus of numerous papers [e.g., 14, 1, 15, 16, 2]. Flat minima, around which the value of the loss function is fairly constant, are believed to correspond to configurations where the network generalizes well to the unseen data while the opposite is believed for sharp minima. Following the underlying idea of these propositions, we believe that studying the geometry of the loss landscape

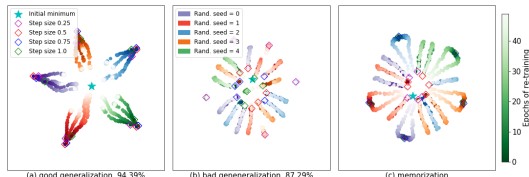

Figure 2: 2D PHATE embeddings of the jump and retrain experiment conducted on one good (**a**) and one bad (**b**) generalization minima as well as one memorization minimum (**c**). The initial points of each retraining are marked by diamonds of different colors based on $step\_size$. All trajectories are colored by their respective direction of the initial jump and descending hue, meaning the color gets whiter as retraining progresses. See Appendix B for a higher resolution version.

around minima has the potential of uncovering key characteristics about the network's ability to generalize. Here, we show that jump-and-retrain experiments combined with our visualization method reveal stark differences between neural network configurations that generalize well versus ones that memorize (or overfit). We trained multiple WResNet28-2 networks on CIFAR10 with and without scrambled labels to obtain generalization and memorization minima. Varying the training procedure for the generalization task allowed us to find "good" and "bad" minima each reaching ~95% and ~85% accuracy on the test set respectively, for more details see Appendix E.

While the starting points are all set on five distinct directions from the minimum, the plotted retraining trajectories are not restricted to any subspace of the parameter space. Despite having only a ~7%

difference in their accuracy on the test data set, the PHATE embedding of the trajectories surrounding each *generalization minimum* is significantly different. The minimum that shows good generalization has a distinctive flower-petal convergence pattern indicating that even when points are thrown far from the minimum, the landscape is wide enough that they return to it in roughly the same direction they were pushed out. This structure is evocative of a high-dimensional "wide valley". In the bad minimum, the trajectories start off near the middle of the plot, but during the retraining the trajectories diverge toward the edges before coming back towards the middle of the plot. This outward movement is stark contrast with the consistent retraining trajectories surrounding the good gen. minimum. This indicates that the region around the bad gen. minimum contains a non-negligible amount of non-convexities, preventing training trajectories to systematically converge towards the minimum. The "memorization" minimum plot Fig. 2c, f looks similar to the bad generalization plot, with trajectories that go outward at small step sizes of the "jump". However, curiously at larger step sizes of the "jump," the trajectories seem to return without going outward first, but they do not return immediately, i.e., they show some lateral movement, potentially indicating bumpiness in the landscape that they are avoiding. See supplemental video for an animation of these trajectories, and Appendix B for trajectories colored by loss. Our method is robust across initializations and tasks since same patterns arise for experiments ran on CIFAR100 and for multiple initializations of the network on CIFAR10, see Appendix F for figures.

# 6 Optimization

We analyzed three stochastic gradient-descent optimization methods commonly used for training ANNs: SGD (vanilla), SGD with momentum (SGD_M), and Adam. The major difference is that Adam uses adaptive step sizes based off previous iterations while SGD and SGD_M use fixed step sizes. First we examine individual trajectories of network weights from the same seed using different optimizers (Fig. 3A). Compared to the smooth trajectory from using Adam, SGD with or without momentum seems to exhibit scattered patterns. It should be noted that the magnitudes of network weights learned by Adam are much larger than by SGD or SGD_M (Appendix C). The evolution and resulting network weights from the Adam optimizer appears to differ more significantly from SGD or SGD_M, where the resulting parameters appear to be further away from the initialization.

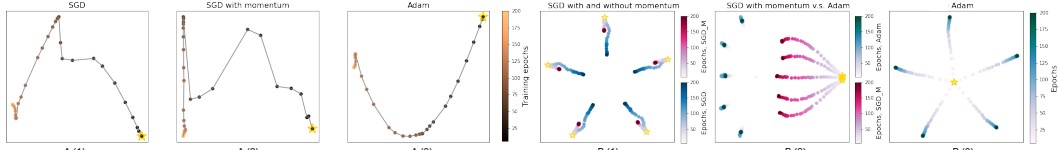

Figure 3: **A** Comparing PHATE embedding of network weights during training using different optimizers. The same seed was used across optimizers, Adam produces a more smooth trajectory. **B** PHATE embedding of network parameters during training using different optimizers (SGD, SGD_M, and Adam). The five initialization are marked by golden stars. B(1),(2) Different optimization algorithms reach distinct minima that all lie in a disc-shaped region. Adam seems to search an orthogonal space of the loss landscape from SGD_M, as seen by the "disappearance" of the trajectory, B(3) The Adam trajectories, plotted by themselves are continuous, however distances between final weights reached by Adam are much larger than other optimizers. See Appendix D for higher resolution.

In Fig. 3B the distinctiveness between optima reached from different initializations and optimizers are clearly shown. We see in the Fig. 3B(1) that when starting from the same seed, SGD and SGD_M travel in similar directions. However, we see a shorter and straighter training path with momentum. Without momentum, a similar minima is reached. However, the path towards the minima is longer with more moves sideways, indicating that the space is searched less effectively. In Fig. 3B(2) the fact that Adam travels so far in the weight space, causes the initial points look very close together. From the initial points, the SGD_M travels in shorter paths towards different minima. However, Adam seems to exhibit a disconnected jump, potentially flying off into an orthogonal space to what is visualized, but interestingly returning to the same region of minima as SGD_M. The seemingly discontinuity of the trajectories in the middle plots indicates that Adam searches space very differently than SGD. Fig. 3B(3) shows that the Adam trajectories are indeed connected in a continuous line, so the trajectory itself is not discontinuous, but rather in a different subspace than the SGD_M trajectory.

# 7 Discussion and conclusion

We proposed a novel approach to visualize ANN loss landscapes based on the state-of-the-art PHATE dimensionality reduction, which is able to capture branch-like structures in high-dimensional data in two dimensional representations by leveraging diffusion-based manifold learning and information geometry. Our approach enables geometric exploration of retrained trajectories surrounding generalization and memorization optima, found via ANN training, to provide insight into generalization capabilities of the network. Further, it enables meaningful comparison of training trajectories across optimizers. Finally, our proposed paradigm is demonstrably consistent and robust across initializations and supervised learning tasks. We expect in future work this visualization approach to enable more methodical paradigms for the development of deep models that generalize better, train faster, and provide fundamental understanding of their capabilities.

# 8 Acknowledgments

This work was partially funded by NSERC's CGS M scholarship [*S.H.*]; IVADO Professor startup & operational funds [*G.W.*]; CZI grants 182702 & CZF2019-002440 [*S.K.*]; and NIH grants R01GM135929 & R01GM130847 [*G.W., S.K.*]. The content provided here is solely the responsibility of the authors and does not necessarily represent the official views of the funding agencies.

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

# Appendix

## A

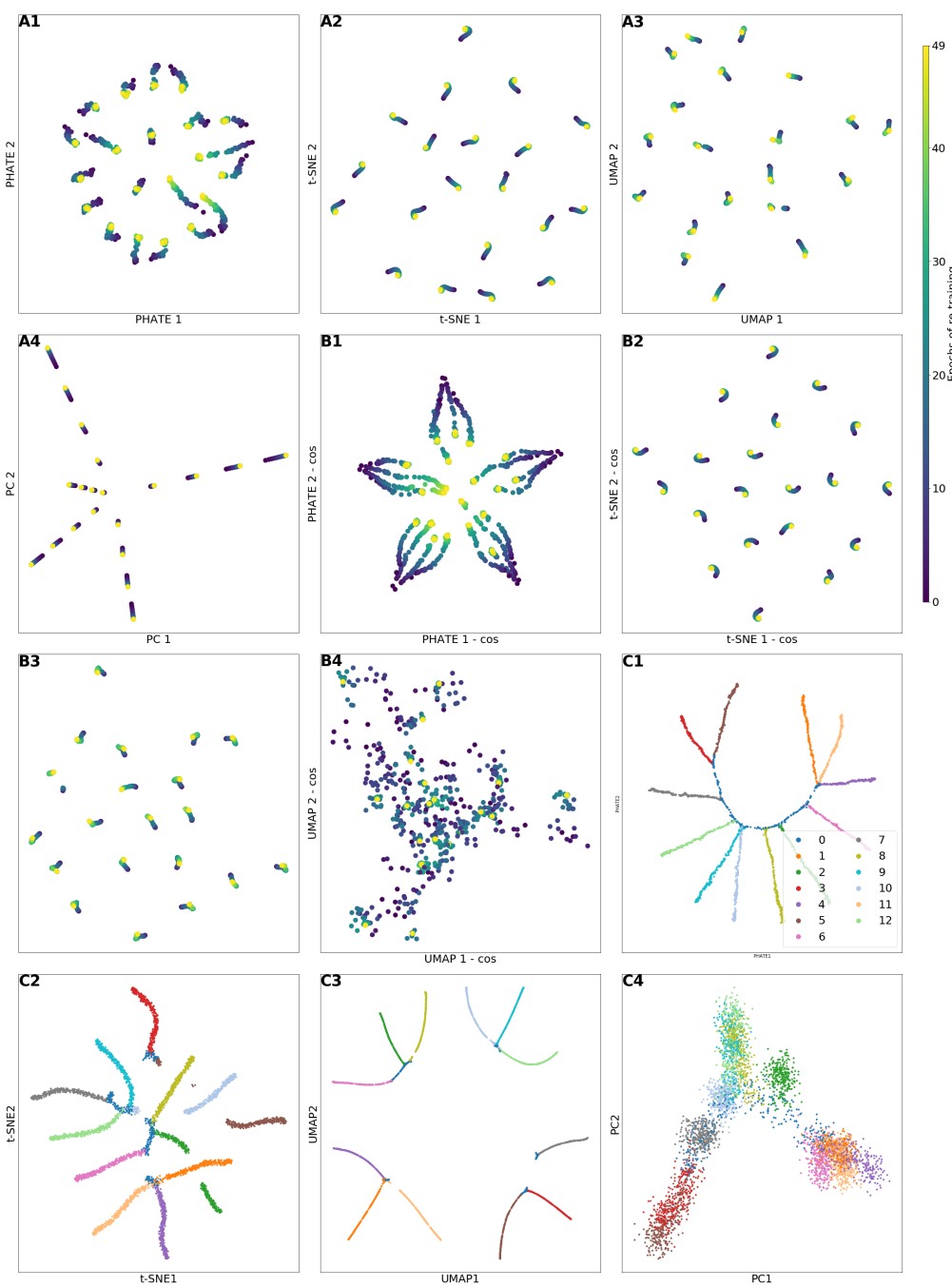

Figure 4: Higher resolution version of Figure 1 with added experiments. 2D embeddings of the jump and retrain experiment results for a WResNet28-2 network using the modern dimensionality reduction techniques PHATE (**1**), t-SNE (**2**), UMAP (**3**) using euclidean (**A**) and cosine (**B**) distances and the linear methods PCA (**A4**) and random plane projection (**B4**). **C:** embeddings found using the same techniques of an artificial data set having a fully connected tree-like structure. We see that PHATE consistently retains continuous trajectory structures while other embeddings (tSNE/UMAP) shatter the structure, or show chaotic trends because of uninformative projections to low dimensions (PCA, random directions). See Appendix A for higher resolution.

**B**

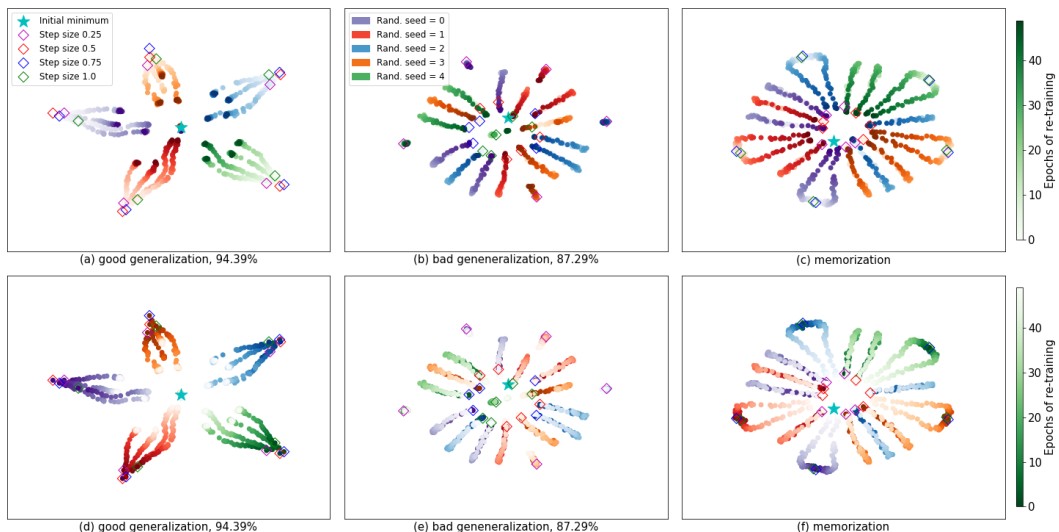

Figure 5: 2D PHATE embeddings of the jump and retrain experiment conducted on one good (**a,d**) and one bad (**b,e**) generalization minima that achieve 94.39% and 87.29% accuracy on the test data respectively, as well as one memorization minimum (**c,f**). The initial points of the jumps (before retraining) are marked by diamonds of different colors based on $step\_size$ in all plots. All trajectories are colored by their respective direction of the initial jump. Plots **a,b,c** have ascending hue, meaning the color gets darker as retraining progresses. Plots **d,e,f** have descending hue, meaning the color gets whiter as retraining progresses. In **d** and **e** the points are colored by the direction ($seed$) of the jumps in parameter space and the color grows darker as retraining progresses. In contrast to more continuous trajectories returning to near the optimum in the good generalization case, memorization displays a more random pattern where weights move out before moving back, often switching direction during retraining. Bad generalization appears to be more similar to memorization, where no continuous trajectories were observed.

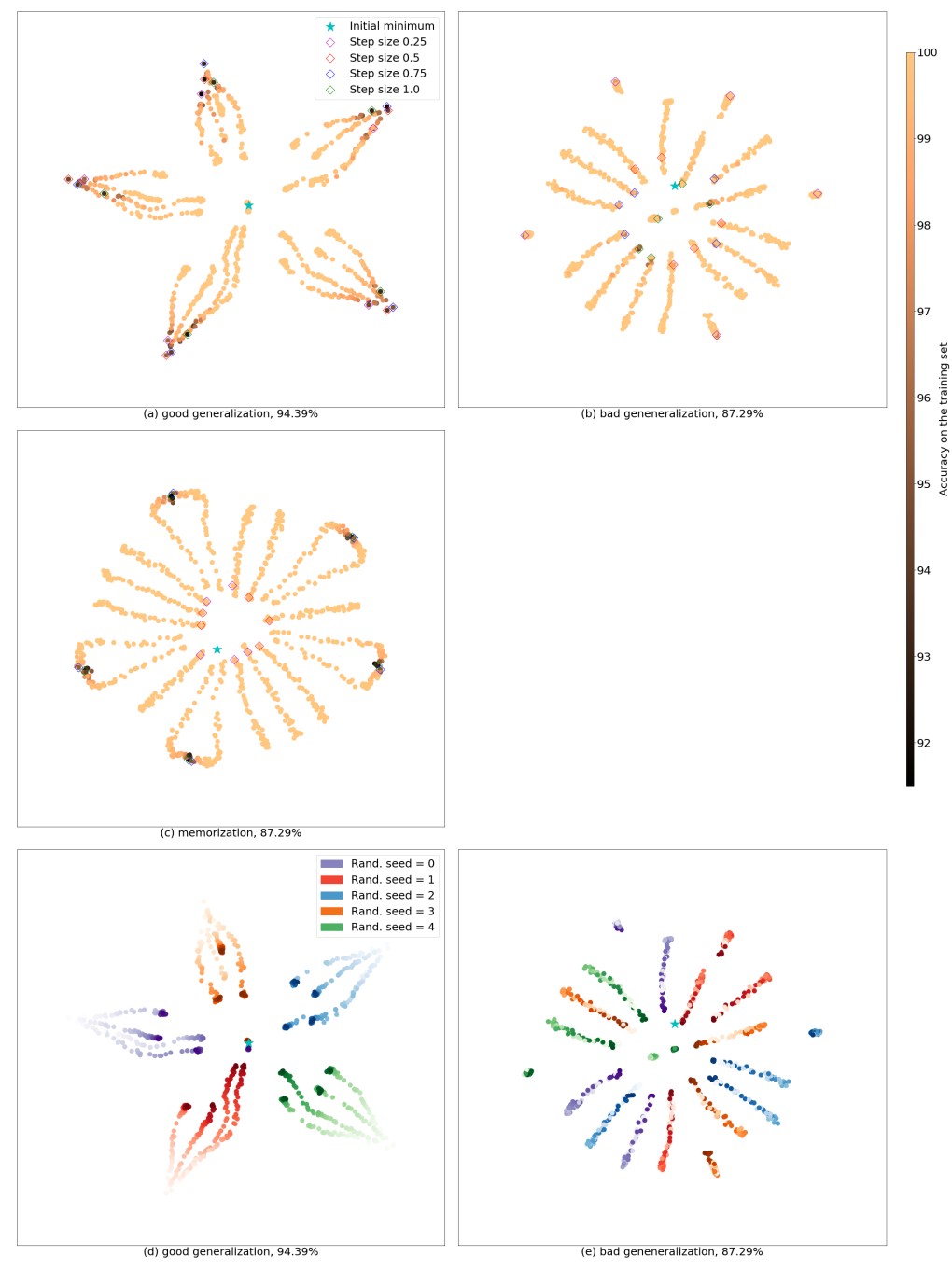

Figure 6: Higher resolution version of Figure 2 but colored according to the loss on the training data set

**C**

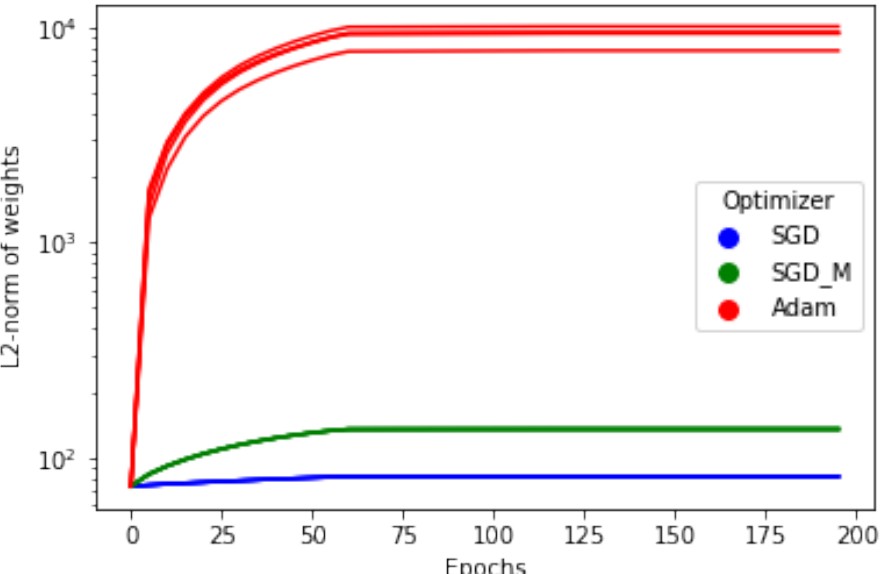

Figure 7: L2 norm of network weights during training using different optimizers.

**D**

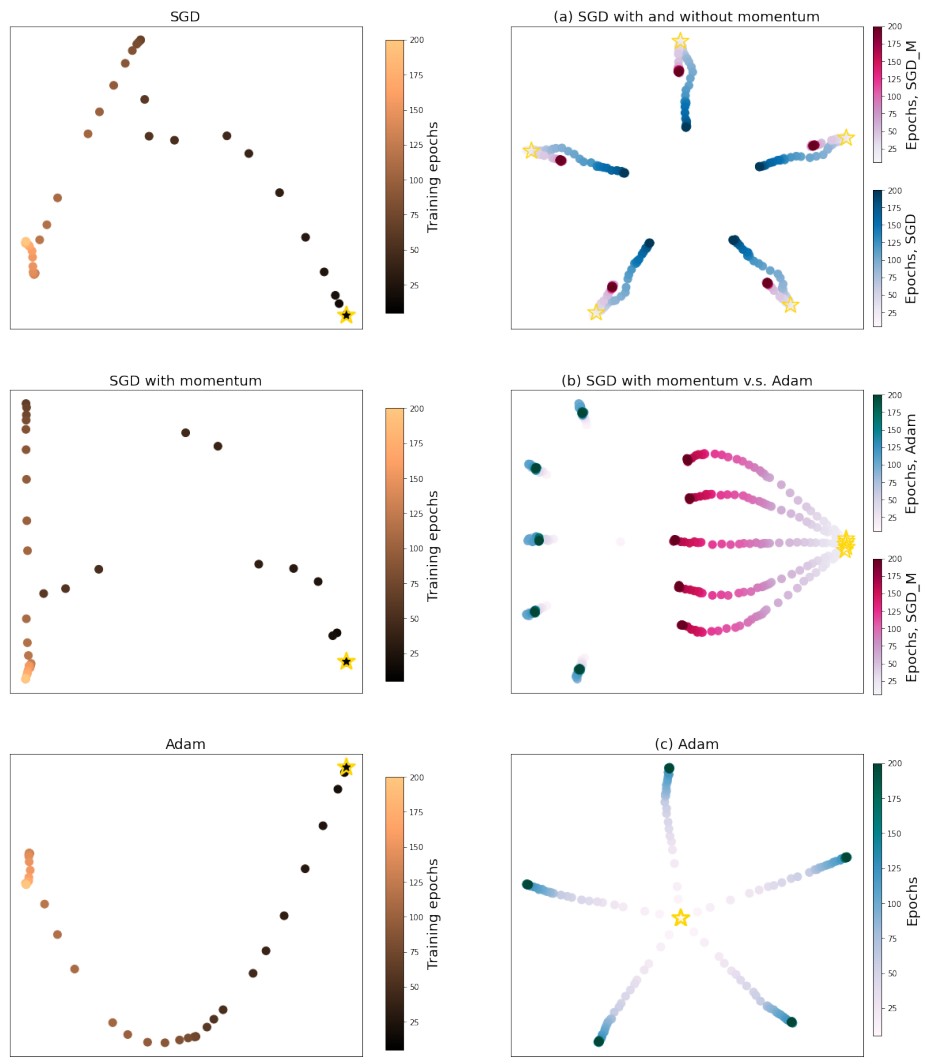

Figure 8: Higer resolution of Figure **??**and Figure **??**

# E   Experimental details

For our empirical results of Section 5 we trained multiple WResNet28-2 networks (wide ResNet of depth 28 and 2 times as wide as a standard ResNet [17]) presented in [18] on the CIFAR10 dataset [19] using batches of size 128 for 200 epochs with the SGD optimizer with 0.9 momentum and a learning rate of 0.1 which decays by a factor of 10 at epoch 150. In some cases we did not use data augmentation and weight decay and in others we used 1e-4 weight decay and random crops and horizontal flips for the training data. Adding the data augmentation and the weight decay has allowed us to find optima that generalize better, increasing the accuracy on the test data from around 85% to around 95%. We will be referring the optima with ~95% test accuracy as "good optima" since they generalize better, and that with lower test accuracy as "bad optima". Using the same training procedure as the one for the bad optima, we also trained a WResNet28-2 network for memorization by completely randomizing the labels of the training data. This modification was used in [20, 21, 22] to show that neural networks have the ability to completely memorize the training data sets. We then ran the jump-and-retrain experiment for these three types of minima and visualized the results using PHATE in Figure 2. This allows us to analyze and compare the regions in parameter space around the good, bad and memorization minima, which all achieve ~0 loss on their respective training sets.

For the results of Section 6 we trained multiple Wide-ResNet networks of depth 28 and twice as wide, similar to the ones in the generali. The same initial learning rate of 0.1 and learning rate decay schedule is employed with each optimizer, data augmentation is used to assist training while no weight decay was used. All experiments achieve close to $100\%$ training accuracy and $> 90\%$ test accuracy. A fixed set of five seeds is used across optimizers.

# F   Robustness

One of the main problems encountered in previous attempts to visualize ANN loss landscapes resides in the difficulty of comparing them across tasks and initializations. Here, we show that our visualization approach, applied to the results of jump-and-retrain experiments, alleviates this concern. To this end, we trained multiple WResNet28-2 networks by following the procedure described in Sec. 2, and starting from multiple parameter initializations. For each initialization, we found one good (~94% test acc.) and one bad optima (~86% test acc.), giving the jump-and-retrain results shown in Fig. 9. Further, we also trained a similar network on the CIFAR100 dataset, following the same training procedure, except here the data augmentation was comprised not only of random horizontal flips and crops, but also of random rotations of up to 45° in each direction and random modifications to the brightness, contrast, saturation and hue of the images. Since CIFAR100 poses a significantly harder classification task than CIFAR10, our networks only reaches ~60% test accuracy for the bad optimum trained without data augmentation and weight decay, and ~70% test accuracy for the good optimum. The fact that the networks only reached 70% accuracy is just a reflection of the complexity of the task, with better accuracy requiring a better fine-tuning of the architecture and the training procedure. To this end we trained multiple networks by following the procedure described in Sec.5 on CIFAR10 and CIFAR100 to find good and bad generalisation minima. The results of these experiments are shown in Fig. 9. For details see Appendix E.

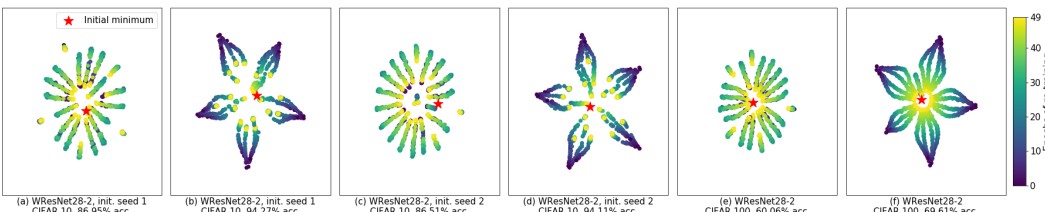

Figure 9: 2D PHATE embeddings of the jump and retrain experiment ran on different initialization for the WResNet28-2 network on CIFAR10 (**a,b,c and d**) and on CIFAR 100 (**e and f**), colored by epoch of retraining. Patterns that resemble bad generalization are exhibited in the first plot for all three settings as in (a), (c) and (e), while good generalization patterns are shown in the second plots (b), (d) and (f). It demonstrates that our approach is robust across tasks. See below for a higher resolution version.

The plots for different initializations of the WResNet28-2 are consistently showing the same star-shaped patterns, characterizing the retraining trajectories surrounding minima that generalize well. The plots for the bad minima also share a similar pattern, not only across initialization but also across tasks. Strikingly, the star-shaped pattern is even better defined in Fig. 9f, corresponding to the good optimum of WResNet28-2 trained on CIFAR100. This offers a key insight into the effect of regularization on the geometric characteristics of the found minima. It seems that regularization techniques enhance generalization by forcing the optimizer into wider valleys.

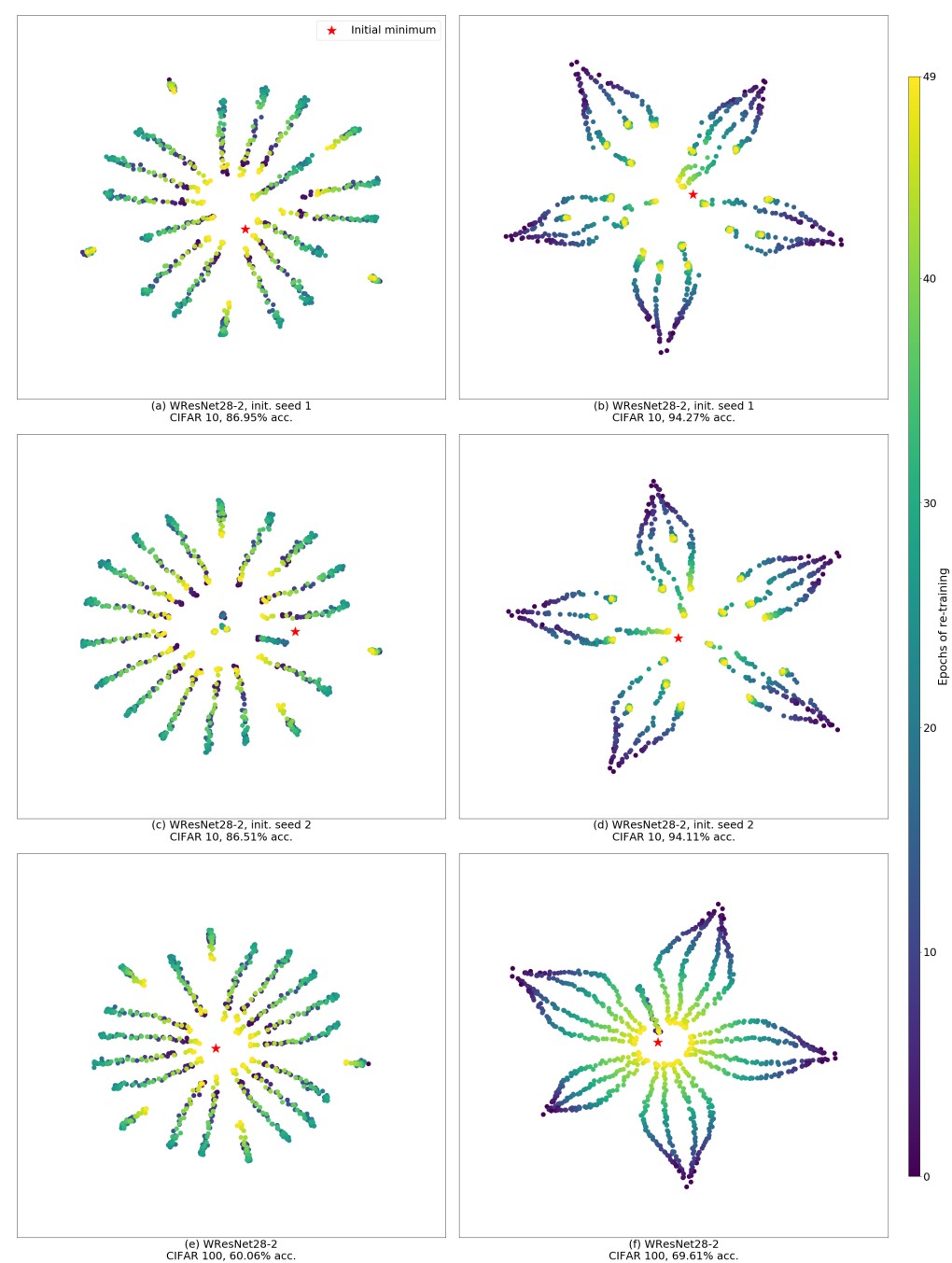

Figure 10: Higher resolution version of Figure 9

