# OpenReview forum: "Visualizing High-Dimensional Trajectories on the Loss-Landscape of ANNs"
_NeurIPS.cc/2020/Workshop/DL-IG — NeurIPSW 2020: DL-IG Poster_

### Official Review · AnonReviewer2 · 2020-10-31
**Review of "Visualizing High-Dimensional Trajectories on the Loss-Landscape of ANNs"**

**Rating:** 9
**Confidence:** 5

**Review:**

The paper applies a high-dimensional data visualization technique called PHATE that was originally developed by computational biologists to visualize biological data.
The authors rightly point out two major problems with the certain recently developed visualization techniques, i.e., they only choose directions that are linear combinations of parameter axes even though the loss landscape itself is clearly highly nonlinear, and that they choose only two among millions of dimensions.

The authors find that the minima exhibiting better generalization have a distinct (and rather curious) flower-petal pattern indicating that the minima are wide. With the help of this visualization method, they also investigate trajectories that various optimizers track.

The paper is well-written with a potentially useful visualization technique. I would point out another method (called disconnectivity diagrams [1]) developed by theoretical chemists/physicists to visualize high-dimensional optimization landscapes can be a more direct comparison to PHATE as this method visualizes the landscape considering all the dimensions at once (compressed to 2D) as opposed to the methods contested against in the current paper. Perhaps both these methods can leverage on each other’s strengths.

[1] Energy Landscapes for Machine Learning. AJ Ballard, et al. Physical Chemistry Chemical Physics 19 (20), 12585-12603 (2017).

---

### Official Review · AnonReviewer1 · 2020-11-01
**review 1**

**Rating:** 7
**Confidence:** 5

**Review:**

This paper visualizes the energy landscape of deep networks using PHATE, which is a nonlinear dimensionality reduction method. The paper argues that this leads to refined understanding of the global structure of the energy landscape as opposed to local methods like PCA, or others like t-SNE and UMAP. The main result is that for weight vectors that generalize well, PHAT results in “flower-like” pattern that indicates wide regions. I find the results for visualizing the trajectory with momentum in Fig. 3 very interesting. The noise in SGD for typical datasets is non-zero only for very few directions in the parameter space, this suggests that momentum with SGD should lead to accelerated convergence and Fig. 3 paints a very clear picture of this phenomenon.

I am a bit concerned about the center panel in Fig. 2. If the trajectories for weights that do not generalize well “go out” from the initialization before coming back, does this mean that the loss increases and then decreases along these trajectories? It would be very useful to color the scatter plot using the training/validation loss as well.

It would be very useful to use these methods to visualize the entire region in the vicinity of a wide region/minimum. This will give us insight into the shape of the minima or, effectively, the Bayes posterior distribution on the weights after training a deep network using SGD.

It would also be useful to visualize datasets such as MNIST/CIFAR-10 using PHAT.

---

### Author Response · Authors · 2020-11-18
**A thank you note for our reviewers**

We thank the reviewers for their thoughtful comments and suggestions. Their constructive criticism will be taken into consideration and will help us improve future iterations of this work.

---

### Author Response · Authors · 2020-12-14
**Video and slides**

Here is a video of the talk presented during the NeurIPS 2020 : DL-IG poster session. The slides are available in the video's description.

https://youtu.be/BRqfSyJe8kU

---

### Decision · Program_Chairs · 2020-11-07

Accept (Poster)